# Do Blood Eosinophils Predict in-Hospital Mortality or Severity of Disease in SARS-CoV-2 Infection? A Retrospective Multicenter Study

**DOI:** 10.3390/microorganisms9020334

**Published:** 2021-02-08

**Authors:** Pierrick Le Borgne, Laure Abensur Vuillaume, Karine Alamé, François Lefebvre, Sylvie Chabrier, Lise Bérard, Pauline Haessler, Stéphane Gennai, Pascal Bilbault, Charles-Eric Lavoignet

**Affiliations:** 1Emergency Department, Hôpitaux Universitaires de Strasbourg, 67091 Strasbourg, France; karine.alame@chru-strasbourg.fr (K.A.); sylvie.chabrier@chru-strasbourg.fr (S.C.); pascal.bilbault@chru-strasbourg.fr (P.B.); 2INSERM (French National Institute of Health and Medical Research), UMR 1260, Regenerative NanoMedicine (RNM), Fédération de Médecine Translationnelle (FMTS), University of Strasbourg, 67081 Strasbourg, France; 3CREMS Network (Clinical Research in Emergency Medicine and Sepsis), 67022 Wolfisheim, France; l.abensurvuillaume@chr-metz-thionville.fr (L.A.V.); charles-eric.lavoignet@hnfc.fr (C.-E.L.); 4Emergency Department, Regional Hospital of Metz-Thionville, 57530 Ars-Laquenexy, France; 5Department of Public Health, University Hospital of Strasbourg, 67091 Strasbourg, France; francois.lefebvre@chru-strasbourg.fr; 6Emergency Department, Haguenau Hospital, 67500 Haguenau, France; lise.berard@ch-haguenau.fr; 7Emergency Department, Colmar Hospital, 68000 Colmar, France; pauline.haessler@ch-colmar.fr; 8Emergency Department, Reims University Hospital, 51100 Reims, France; sgennai@chu-reims.fr; 9Emergency Department, Hôpital Nord Franche Comté, 90400 Trévenans, France

**Keywords:** COVID-19, eosinophils, eosinopenia, severity, mortality

## Abstract

Introduction: Healthcare systems worldwide have been battling the ongoing COVID-19 pandemic. Eosinophils are multifunctional leukocytes implicated in the pathogenesis of several inflammatory processes including viral infections. We focus our study on the prognostic value of eosinopenia as a marker of disease severity and mortality in COVID-19 patients. Methods: Between 1 March and 30 April 2020, we conducted a multicenter and retrospective study on a cohort of COVID-19 patients (moderate or severe disease) who were hospitalized after presenting to the emergency department (ED). We led our study in six major hospitals of northeast France, one of the outbreak’s epicenters in Europe. Results: We have collected data from 1035 patients, with a confirmed diagnosis of COVID-19. More than three quarters of them (76.2%) presented a moderate form of the disease, while the remaining quarter (23.8%) presented a severe form requiring admission to the intensive care unit (ICU). Mean circulating eosinophils rate, at admission, varied according to disease severity (*p* < 0.001), yet it did not differ between survivors and non-survivors (*p* = 0.306). Extreme eosinopenia (=0/mm^3^) was predictive of severity (aOR = 1.77, *p* = 0.009); however, it was not predictive of mortality (aOR = 0.892, *p* = 0.696). The areas under the Receiver operating characteristics (ROC) curve were, respectively, 58.5% (CI95%: 55.3–61.7%) and 51.4% (CI95%: 46.8–56.1%) for the ability of circulating eosinophil rates to predict disease severity and mortality. Conclusion: Eosinopenia is very common and often profound in cases of severe acute respiratory syndrome coronavirus 2 (SARS-CoV-2) infection. Eosinopenia was not a useful predictor of mortality; however, undetectable eosinophils (=0/mm^3^) were predictive of disease severity during the initial ED management.

## 1. Introduction

For several months now, medical systems worldwide have been battling an ongoing pandemic caused by a novel coronavirus, SARS-CoV-2 (severe acute respiratory syndrome coronavirus 2) [1]. According to the World Health organization, as of January 2021, one year after the start of the pandemic, this emerging virus infected almost 100 million people and resulted in nearly 2.2 million deaths around the world. In this global health crisis, emergency departments (ED) have been fighting at the front lines. As with other acute pathologies (such as myocardial infarction, stroke and sepsis), clinical anticipation and accurate triage is crucial. It implies rapid identification of the most critical patients in order to optimize patient management.

Complete blood count (CBC) is an easily accessible and inexpensive routine set of medical laboratory tests. Numerous recent studies have described changes in white blood cell counts of patients with COVID-19, including a significant decrease in eosinophils and circulating lymphocytes [2,3,4,5,6]. The diagnostic and prognostic performance of blood eosinophils has been observed in different pathologies; this is mainly demonstrated in deep eosinopenia, with an eosinophil count below 50 to 10 per mm^3^ [7,8,9,10,11,12]. The prognostic value of eosinopenia is described in bacterial infections presented by critically ill patients [7,8,9,10], and in non-infectious diseases such as myocardial infarction [11] and stroke [12]; thus far, eosinopenia appears to be a marker of increased mortality. 

In light of the ongoing COVID-19 outbreak, many researchers have studied the diagnostic value of eosinophils and its contribution to identify COVID-19 cases. By itself, eosinopenia does not appear to be sufficiently powerful as biomarker for positive diagnosis of SARS-CoV-2 infection [5]. However, in a retrospective study of 989 patients presenting with fever, Li et al. [5] described an area under the curve (AUC) of 0.73 associating eosinopenia below 20/mm^3^ with an increase in high-sensitivity C-reactive protein (hs-CRP). Similarly, Formica et al. [13] identified eosinopenia below 10/mm^3^ as a significant parameter and integrated it in a combined score with a discriminatory power for COVID-19 diagnosis and an AUC of 92%. Moreover, several other teams have studied the prognostic value of eosinopenia, aiming rapid and precise identification of critically ill COVID-19 patients and those at risk of developing severe complications. These studies describe a lower eosinophils rate in the most severe patients [14,15,16], yet the significance of these findings remains controversial, particularly in recent meta-analyses [17,18]. The results, thus far, appear contradictory and the prognostic performance of eosinopenia in SARS-CoV-2 infection remains uncertain. In this work, we focused on studying the prognostic value of eosinopenia in COVID-19 patients upon admission to the ED.

## 2. Methods

### 2.1. Study Population and Settings

This retrospective, multicenter study was conducted in six EDs in northeast France; in two university hospitals (CHRU of Strasbourg and CHU of Reims) and four general hospitals (Colmar Hospital, Nord Franche-Comté Hospital, Metz-Thionville Hospital and Haguenau Hospital). These six centers, along with the entire Greater-East region of France, have been heavily impacted by the pandemic. As of the end of June, this area reported nearly 3500 deaths and 12,000 patients infected with SARS-CoV-2. 

From 1 March to 30 April 2020, all adult patients who were admitted to the ED and were hospitalized for COVID-19 were included in our study. Patients were managed following current guidelines, which, at that time, did not rely on any specific therapeutic intervention. All patients included had at least one nasopharyngeal swab where RT-PCR was positive for SARS-CoV-2. Patients who had no positive swab during their hospital stay and those who received outpatient care were excluded. Patients with a medical history or treatment that altered their blood counts and therefore circulating eosinophils (e.g., chemotherapy, immunosuppressive therapy, long- and short-term corticosteroid therapy, pre-hospital antibiotic therapy, active cancer or hematological malignancies) were also excluded from our study. Those who received palliative therapy or limitation of therapeutic effort upon admission to the ED were also excluded. 

### 2.2. Data Collection

We retrospectively compiled data from patients’ electronic medical records and then standardized it in a report file. The collected data included epidemiological, clinical and biochemical parameters. Symptom onset date was recorded. Patients’ current treatments and medical history, including cardiovascular diseases, diabetes, pre-existing renal failure, cancer and hematological diseases, were also collected. In addition, the autonomy of each patient was recorded using the Knaus score [19]. In this study, the severity of disease was defined by admission to the Intensive Care Unit (ICU). Obesity was defined by a body mass index superior to 30 kg/m^2^. Standard biochemical parameters, such as creatinine, C-reactive protein (CRP), total leukocytes and lymphocytes, were also collected. Concerning circulating eosinophils, their normal rate stands between 100 and 400/mm^3^, eosinopenia is considered profound when the rate is inferior to 50/mm^3^ and undetectable when it equals 0/mm^3^ [7,8,9,10,11,12].

Lastly, we measured eosinophil count variation, delta, the difference between circulating eosinophils rate at admission in the ED, and after 24 hours of ED management (H-24). All collected data are summarized in the results sections.

### 2.3. Ethics

This study was approved by the local ethics committee of the University of Strasbourg in France (reference CE: 2020-39), which, in accordance with the French legislation, waived the need for informed consent of patients whose data were entirely retrospectively studied.

### 2.4. Statistical Analysis

The statistical analyses included a descriptive section and an analytical section. Descriptive analysis of qualitative variables was performed by giving the frequency of each value along with the cumulative frequency. Descriptive analysis of quantitative variables was performed by giving location parameters (mean, median, minimum, maximum, first and third quartiles) and dispersion parameters (standard deviation, variance, range and interquartile range). Normality of the distributions was tested using a normality test, such as the Shapiro–Wilk or Kolmogorov–Smirnov test, and was assessed graphically using a normal quantile plot. Comparisons between qualitative variables were performed using Chi-squared test or Fisher’s exact test in case of expected values in any of the cells of a contingency table are below 5. Comparisons between quantitative and qualitative variables were assessed using the Student’s t-test or Wilcoxon’s test in case of heteroskedasticity or if the variable did not follow a normal distribution. Multivariate analyses were performed using all relevant variables. The significance level was set at 5%. All the statistical analyses were generated with R 4.0.2.

## 3. Results

### 3.1. Clinical Characteristics of the Study Population 

During the study period, which lasted two months, 49,326 patients were addmitted to the EDs of all study centers combined. Of these patients, 4470 were diagnosed with SARS-CoV-2 infection using RT-PCR on nasopharyngeal swab. A total of 1035 patients were included in our study (Figure 1). The majority of patients were male (58.8%, CI95%: 55.8–61.8%), median age was 69 (58–79) years, and over a third of the study population was obese (34%). In terms of medical history, over half of the patients (56.7%, CI95%: 53.7–59.7%) had high blood pressure, over a quarter of them (26.7%) had a history of diabetes, and 23.2% (CI95%: 20.6–25.8%) of them presented pre-existing renal failure. Over three-quarters of the patients (77.2%, CI95%: 74.6–79.8%) did not show any loss of functional autonomy, as measured by the Knaus score. The vast majority of patients (92.8%, CI95%: 91.1–94.3%) presented eosinopenia (below 100/mm^3^) upon admission to the ED, and this decrease in eosinophils was mostly deep (below 50/mm^3^) (87.4%, CI95%: 85.2–89.4%). The remaining characteristics are summarized in Table 1.

### 3.2. Comparison and Correlation according to the Severity of Disease

A total of 789 (76.2%) patients presented moderate disease, whereas 246 (23.8%) patients presented severe disease and were admitted to the ICU. When comparing these two subgroups, age (70 vs. 66 years, *p* < 0.001) and gender (*p* < 0.001) differed significantly. Patients admitted to the ICU had fewer cardiovascular (*p* = 0.004) and renal (*p* = 0.002) comorbidities; in addition, they frequently presented no pre-existing limitation in daily activity (*p* < 0.001). Clinically, the initial assessment upon arrival to the ED showed signs of respiratory failure (lower saturation, increasing oxygen requirement) more frequently in the severe disease subpopulation (*p* < 0.001). Biochemically, upon admission to the ED, CRP (86.3 versus 143.8 mg/L, *p* < 0.001) and lactate (*p* < 0.001) were more elevated in the subgroup presenting severe disease. On the first CBC, mean lymphocyte count did not differ between the two subgroups (*p* = 0.297). Yet, conversely, mean circulating eosinophils rate was lower in patients admitted to the ICU (22.6 versus 11.2/mm^3^, *p* < 0.001). Similarly, on the second CBC, performed 24 h after admission to the ED, mean eosinophils rate was lower in patients admitted to the ICU (*p* < 0.001). Moreover, we measured eosinophil count variation, delta, the difference between circulating eosinophils rate at admission to the ED, and H-24. We found that positive deltas of eosinophils at H-24 were significantly higher in the moderate disease group (*p* < 0.001). Concerning the overall hospital stay, duration of oxygen therapy (5 versus 20 days, *p* < 0.001) and the number of thromboembolic events (*p* < 0.001) were higher in the subgroup presenting severe disease. When admitted to the ICU, duration of hospital stay (8 vs. 24 days, *p* < 0.001) was longer and in-hospital mortality (10.4 vs. 24.1%, *p* < 0.001) was higher. All these results are summarized in Table 1. 

Additionally, we studied the prognostic value of eosinopenia as a predictor of disease severity. The area under the ROC curve was 58.5% (CI95%: 55.3–61.7%) (Figure 2). For the multivariable analysis, factors associated with disease severity were male gender (OR = 1.554, *p* = 0.04) and elevated CRP (over 100 mg/dL) (OR = 2.94, *p* < 0.001). Undetectable blood eosinophils (0/mm^3^) was, in addition, a predictor of disease severity (OR = 1.77, *p* = 0.009). Conversely, age (OR 0.975, *p* < 0.001) and a positive differential of eosinophils at H-24 (positive delta at H-24) (OR = 0.273, *p* < 0.001) were protective factors regarding disease severity. These results are summarized in Table 2. 

### 3.3. Comparison and Correlation according to Survival

Mortality analysis included 1023 patients, as 12 patients (1.2%) were lost to follow-up. In total, 139 (13.6%) patients died during their hospital stay. Non-survivors were significantly older (78 versus 67 years, *p* < 0.001). They were more likely to have a medical history of hypertension (*p* < 0.001) and renal failure (*p* < 0.001). In addition, they presented more limitation in daily activity (*p* < 0.001). Clinically, the initial assessment upon arrival to the ED showed signs of respiratory failure (lower saturation, increasing oxygen requirement) more frequently in the non-survivor subgroup (*p* < 0.001). Biochemically, higher levels of creatinine (120 vs. 89 μmol/L, *p* < 0.001), CRP (116 vs. 96 mg/L, *p* < 0.001), and lactate (*p* = 0.001) were observed in the non-survivor subgroup. Regarding the CBC, lymphopenia was more profound (*p* = 0.016), while neutrophil counts were higher (*p* = 0.035) in the non-survivor subgroup. As for circulating eosinophil counts, mean rate at admission did not differ between the two subgroups (*p* = 0.306), yet, at H-24 (*p* = 0.005), eosinophil counts were higher in the surviving subgroup. Concerning the remainder of the hospital stay, the non-survivors received more antibiotics (*p* = 0.019) and longer oxygen therapy (*p* < 0.001). The length of hospital stay did not vary between the two subgroups (*p* = 0.064). These results are summarized in Table 3. 

Additionally, we studied the prognostic value of eosinopenia as a predictor of disease mortality. The area under the ROC curve was 51.4% (CI95%: 46.8–56.1%) (Figure 3). On multivariable analysis, the factors associated with mortality were age (OR = 1.056, *p* < 0.001), obesity (OR = 1.848, *p* = 0.045), and male gender (OR = 2.043, *p* = 0.019). All other co-morbidities and biological parameters studied showed no significant association. Hence, undetectable eosinopenia (=0/mm^3^) was not predictive of mortality (OR = 1.006, *p* = 0.982). Similarly, a positive eosinophil delta between admission and H-24 was not predictive of mortality (OR = 0.696, *p* = 0.218). All these results are shown in Table 4.

## 4. Discussion

Our main objective was to study the prognostic value of eosinopenia in patients admitted to the ED, then hospitalized for SARS-CoV-2 infection. A large majority of patients in our cohort had deep eosinopenia (below 50/mm^3^). In a retrospective study of 48 patients with COVID-19, Song et al. [20] also found eosinopenia (below 40/mm^3^) in 75% of patients. These findings are also consistent with a recent meta-analysis [18]. Concerning mortality, our study showed no significant difference in circulating eosinophils rate at admission when comparing survivors to non-survivors; tendency measurements even showed a slightly higher rate in non-survivors (24.4 vs. 19/mm^3^, *p* = 0.306). The area under the ROC curve was very low. 

Regarding clinical severity, we managed to demonstrate a significant difference in eosinophil counts when comparing moderate and severe disease forms, yet this difference is relatively modest with, respectively, 22.6 versus 11.2 eosinophils/mm^3^ (*p* < 0.001). Although significant, this difference seems difficult to interpret and incorporate in current clinical practices as a severity predictor. Commonly used automated cell counters detect cells with a sensitivity that varies between 1 to 10 eosinophils/mm^3^, which puts us at risk of falsely disregarding minor differences. Here, again, the area under the ROC curve was very low. In addition, Rocca et al. [17] found, in a cohort of 294 patients, a modest and weakly significant difference in circulating eosinophils rate, with 32 eosinophils/mm^3^ in moderately affected patients and 19 eosinophils/mm^3^ in severe patients (*p* = 0.049). In the same study, the authors also found a significant difference in circulating eosinophils rate when comparing non-survivors and survivors (22 vs. 31 eosinophils/mm^3^, *p* = 0.032) [17]. Furthermore, in a meta-analysis collecting eosinopenia data from five studies and comparing patients with severe disease to those with moderate form, Gahramani et al. [18] found a weighted mean difference (WMD) of −0.03 × 109/L (95%CI −0.05–0.00 × 109/L) with high heterogeneity (I2 = 86%). In another meta-analysis of four Chinese studies, totaling 347 patients, Henry et al. [21] found a WMD of −0.01 (95IC −0.02–0.0) with high heterogeneity as well (I2 = 74.4%). 

In light of our results and considering all data collected from literature to date, we, therefore, conclude that eosinopenia holds no major interest in clinical practice as an isolated prognostic marker in COVID-19 patients. However, eosinopenia could be coupled with other markers in a multimarker approach or in a combined severity prediction score, increasing the prognostic value of this biomarker in COVID-19. It should be noted that, similarly, numerous other biochemical markers have been described to identify the most severe patients, including lymphopenia, neutrophils, neutrophil to lymphocyte ratio, CRP level, and D-Dimers [18,20,21]. More studies are, thus, required to further explore this prospect. 

Finally, profound and persisting eosinopenia at H-24 might be an interesting predictor of both severity and mortality, as a significant difference was observed in both cases. Other studies described similarities between the dynamics of circulating eosinophils rate and clinical evolution of COVID-19 [22,23]. Mu et al. [22] described increased eosinophil counts in a series of 95 patients with favorable clinical outcome, whereas persistent undetectable eosinophil counts seemed to predict an unfavorable clinical outcome. Thus, persistent eosinopenia at H-24 could be an interesting tool to judge the subsequent evolution of patients with SARS-CoV-2 infection.

The physiopathology of eosinopenia is yet to be fully understood. In addition to their known functions in allergic and anti-parasitic processes, eosinophils are involved in the initiation and propagation of immune response [24]. Various physiological stress responses appear to be accompanied by a decrease in circulating eosinophils rate [25,26,27]. Eosinophil margination seems to be caused by the production of chemotactic factors and complement activation within the inflammatory site [25]. Eosinophils’ anti-viral properties reside in molecules released by degranulation and in the role these cells play in exposing viral antigens [26,27]. Although the anti-viral role of eosinophils has previously been described, it remains controversial, especially when it concerns COVID-19. While the prognostic and diagnostic performance of eosinopenia has been demonstrated, their actual involvement in inflammatory pathways of infectious diseases is yet to be fully established [28,29]. Thus, in SARS-CoV-2 infection, some authors have hypothesized the possibility of eosinophils migration, similar to that of lymphocytes, into the pulmonary parenchyma, leading to a decrease in these circulating white blood cells [29].

### Limitations

Our study presents some limitations. Firstly, in its retrospective nature, yet it should be noted that we led our research project in six major EDs in the northeast region of France, which was one of the pandemic’s epicenters during its first wave. Secondly, we had to exclude an extensive number of patients due to their medical history, notably onco-hematological comorbidities, as it modifies blood counts and, therefore, circulating eosinophils rate. Similarly, many patients were admitted to the ED after receiving non-recommended antibiotics which affect CBC; those were also excluded from the study. Thirdly, due to the absence of a control group (COVID-), we were unable to study the diagnostic performance of eosinophils in SARS-CoV-2 infection. Further studies are required, notably on larger prospective cohorts, and probably as part of a multimarker approach. Lastly, given the overwhelming workload and pressure submerging our healthcare systems during the first wave of the outbreak, overall parameters of the hospital stay could not be exhaustively collected and detailed. 

## 5. Conclusions

Eosinopenia is very common and often profound in cases of SARS-CoV-2 infection. Eosinopenia was not a useful predictor of mortality; however, undetectable eosinophils were predictive of disease severity during the initial ED management.

## Figures and Tables

**Figure 1 microorganisms-09-00334-f001:**
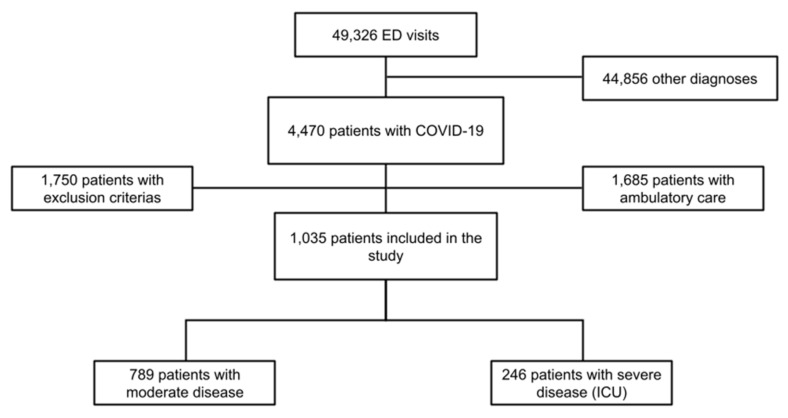
Flowchart of the study. Abbreviations: ED = Emergency Department, ICU = intensive care unit, COVID-19 = coronavirus disease 2019.

**Figure 2 microorganisms-09-00334-f002:**
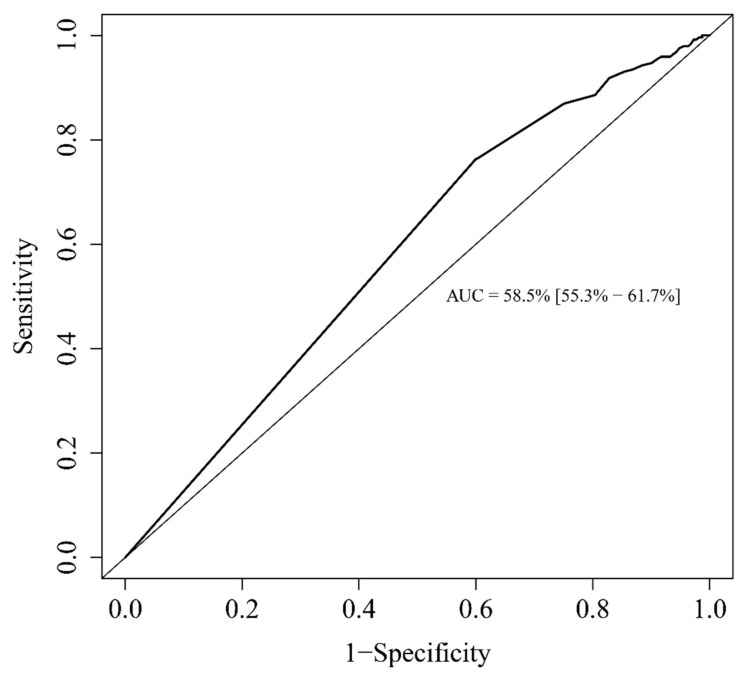
Receiver operating characteristics (ROC) curve for the ability of circulating eosinophils to predict severity of disease (ICU admission).

**Figure 3 microorganisms-09-00334-f003:**
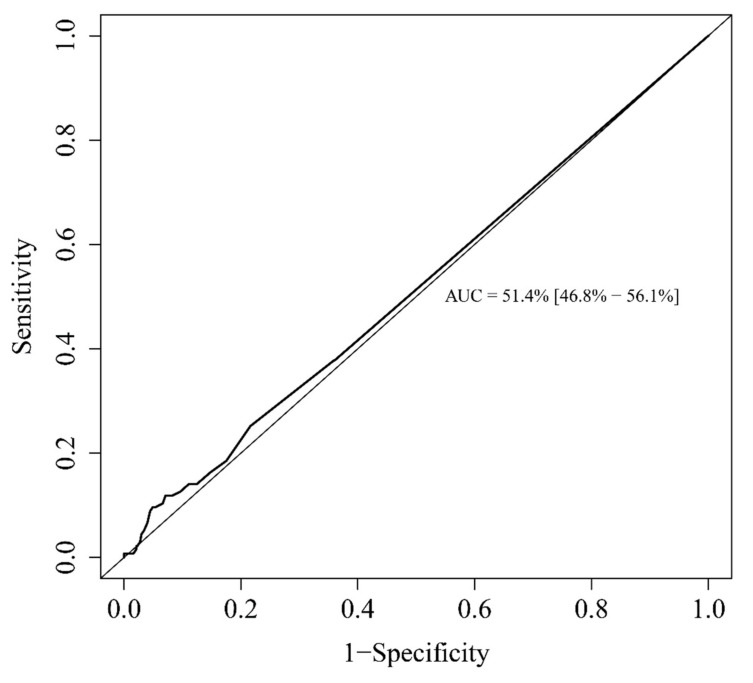
Receiver operating characteristics (ROC) curve for the ability of eosinophils to predict death. Abbreviation: ICU = intensive care unit.

**Table 1 microorganisms-09-00334-t001:** General characteristics of study population with severe acute respiratory syndrome coronavirus 2 (SARS-CoV-2) infection and comparison according to disease severity.

	Total Patients *n* = 1035	Moderate Severity *n* = 789	Severe (ICU) Patients *n* = 246	*p* Value
**Demographics**				
Age	69.0 [58.0–79.0]	70.0 [58.0–81.0]	66.0 [57.3–72.0]	<0.001 *
Gender (male)	609 (58.8)	433 (54.9)	176 (71.5)	<0.001 *
Obesity (BMI > 30)	259 (34.0)	178 (32.3)	81 (38.6)	0.100
**Comorbidities**				
Hypertension	587 (56.7)	453 (57.4)	134 (54.5)	0.416
Diabetes mellitus	275 (26.6)	202 (25.6)	73 (29.7)	0.207
Pre-existing renal failure	237 (23.2)	199 (25.5)	38 (15.8)	0.002 *
Cardio-vascular diseases	357 (34.5)	291 (36.9)	66 (26.8)	0.004 *
Total autonomy	796 (77.2)	569 (72.4)	227 (92.7)	<0.001 *
**Presentation in the ED**				
Saturation with O_2_ (%)	92.0 [88.0–95.0]	93.0 [89.0–96.0]	88.0 [80.0–91.0]	<0.001 *
O_2_ requirement (L/min)	2.0 [0.0–4.0]	2.0 [0.0–3.0]	4.5 [2.0–9.3]	<0.001 *
Time form symptom onset (days)	7.0 [3.0–9.0]	6.0 [3.0–9.0]	7.0 [4.0–9.0]	0.014 *
**Laboratory findings**				
Creatinine (μmol/L)	93.4 ± 70.9	93.3 ± 77.4	93.9 ± 44.2	0.882
C-reactive protein (mg/L)	99.9 ± 79.4	86.3 ± 68.8	143.8 ± 94.3	<0.001 *
Lactate (mmol/L)	1.4 ± 0.9	1.3 ± 0.9	1.6 ± 1.0	<0.001 *
Lymphocytes (/mm^3^)	969.1 ± 540.4	981.1 ± 478.7	930.9 ± 701.2	0.297
Neutrophils (/mm^3^)	5617.7 ± 3261.8	5390.7 ± 3059.7	6335.8 ± 3749.1	<0.001 *
Eosinophils (/mm^3^)	19.9 ± 48.7	22.6 ± 52.3	11.2 ± 33.7	<0.001 *
Eosinophils H-24 (/mm^3^)	38.3 ± 78.6	44.0 ± 84.9	22.4 ± 54.4	<0.001 *
Delta eosino + (H-24)	352 (43.2)	293 (48.8)	59 (27.6)	<0.001 *
**Hospital stay**				
Antibiotics	457 (44.2)	336 (42.6)	121 (49.4)	0.061
O_2_ requirement (days)	7.0 [3.0–13.0]	5.0 [1.0–8.0]	20.0 [13.0–30.0]	<0.001 *
**Outcome**				
Thrombo-embolic events	68 (6.6)	26 (3.3)	42 (17.4)	<0.001 *
In hospital LOS (days)	10.0 [7.0–17.3]	8.0 [6.0–12.0]	24.0 [17.0–38.0]	<0.001 *
In hospital mortality	139 (13.6)	82 (10.4)	57 (24.1)	<0.001 *

Data are all expressed in median [Q1–Q3] or mean ± SD or *n* (%) where *n* is the total number of patients with available data. * *p* < 0.05. Abbreviations: BMI = body mass index, ED = Emergency Department, O_2_ = oxygen, Eosino = eosinophils Delta+ = positive difference between circulating eosinophils rate at admission to the ED, and after 24 h of hospital stay (H-24), LOS = length of stay.

**Table 2 microorganisms-09-00334-t002:** Multivariable analysis of patients with SARS-CoV-2 infection according to disease severity (ICU admission).

Characteristics	Odds Ratio	95%CI	*p* Value
Age	0.975	0.959 0.990	0.001 *
Obesity (BMI > 30)	1.266	0.832 1.927	0.271
Gender (men)	1.554	1.018 2.374	0.041 *
**Comorbidities**			
Pre-existing renal failure	0.840	0.494 1.428	0.519
Hypertension	1.071	0.665 1.725	0.777
Diabetes mellitus	1.082	0.684 1.712	0.735
**Laboratory findings**			
Creatinine > 100 μmol/L	1.339	0.813 2.206	0.251
CRP > 100 mg/L	2.941	1.946 4.447	<0.001 *
Lymphopenia < 500/mm^3^	1.008	0.584 1.739	0.978
Neutrophils > 10000/mm^3^	1.658	0.871 3.158	0.124
Eosinophils = 0/mm^3^	1.769	1.152 2.717	0.009 *
Delta Eosinophils + (H-24)	0.273	0.178 0.418	<0.001 *

Abbreviations: ICU = intensive care unit, BMI = body mass index, CRP = C reactive protein, Delta + = positive difference between circulating eosinophils rate at admission to the ED, and after 24 hours of hospital stay (H-24). * *p* < 0.05.

**Table 3 microorganisms-09-00334-t003:** Overall characteristics of patients with SARS-CoV-2 and comparison according to survival.

	Total Patients *n* = 1023	Survivors *n* = 884	Non-Survivors *n* = 139	*p* Value
**Demographics**				
Age	69.0 [58.0–79.0]	67.0 [56.0–77.0]	78.0 [70.0–86.0]	<0.001 *
Gender (male)	602 (58.9)	517 (58.5)	85 (61.2)	0.553
Obesity (BMI > 30)	258 (34.1)	227 (33.9)	31 (36.1)	0.690
**Comorbidities**				
Hypertension	580 (56.7)	477 (54.0)	103 (74.1)	<0.001 *
Diabetes mellitus	269 (26.3)	227 (25.7)	42 (30.2)	0.259
Pre-existing renal failure	236 (23.4)	189 (21.6)	47 (35.3)	<0.001 *
Cardio-vascular Disease	355 (34.7)	283 (32.1)	72 (51.8)	<0.001 *
Total autonomy	785 (77.0)	702 (79.6)	83 (60.6)	<0.001 *
**Presentation ED**				
Saturation with O_2_ (%)	92.0 [88.0–95.0]	92.0 [88.0–95.0]	88.0 [80.0–93.0]	<0.001 *
O_2_ requirement (L/min)	2.0 [0.0; 4.0]	2.0 [0.0–3.0]	3.0 [2.0–6.0]	<0.001 *
Time form symptom onset(days)	7.0 [3.0; 9.0]	7.0 [3.0–9.0]	3.0 [2.0–7.0]	<0.001 *
**Laboratory findings**				
Creatinine (μmol/L)	93.4 ± 71.1	89.1 ± 71.0	120.3 ± 65.5	<0.001 *
C-reactive protein (mg/L)	99.3 ± 79.1	96.7 ± 77.0	116.6 ± 89.4	0.014 *
Lactate (mmol/L)	1.4 ± 0.9	1.3 ± 0.9	1.7 ± 1.0	0.001 *
Lymphocytes (/mm^3^)	968.2 ± 539.5	986.5 ± 524.6	849.6 ± 616.9	0.016 *
Neutrophils (/mm^3^)	5601.2 ± 3263.1	5505.0 ± 3177.4	6223.5 ± 3724.7	0.035 *
Eosinophils (/mm^3^)	19.8 ± 48.6	19.0 ± 46.9	24.4 ± 58.3	0.306
Eosinophils H-24 (/mm^3^)	38.4 ± 78.8	40.8 ± 81.2	22.2 ± 59.0	0.005 *
Delta eosino + (H-24)	349 (43.4)	315 (45.1)	34 (32.1)	0.012 *
**Hospital stay**				
Antibiotics	450 (44.0)	376 (42.6)	74 (53.2)	0.019 *
O_2_ requirement (days)	7.0 [3.0–13.0]	6.0 [2.0–13.0]	8.0 [5.0–16.0]	<0.001 *
**Outcome**				
Thrombo-embolic events	67 (6.6)	55 (6.2)	12 (8.6)	0.285
In hospital LOS (days)	10.0 [7.0–17.5]	10.0 [7.0–18.0]	9.0 [5.0–16.5]	0.064

Data are all expressed in median [Q1–Q3], mean ± SD, or *n* (%), where N is the total number of patients with available data. * *p* < 0.05. Abbreviations: BMI = body mass index, ED = Emergency Department, O_2_ = oxygen, eosino = eosinophils, Delta+ = positive difference between circulating eosinophils rate at admission in the ED, and after 24 hours of the hospital stay (H-24).

**Table 4 microorganisms-09-00334-t004:** Multivariable analysis of factors associated with in-hospital mortality.

Characteristics	Odds Ratio	95%CI	*p* Value
Age	1.046	1.020 1.073	<0.001 *
Obesity (BMI > 30)	1.366	0.740 2.520	0.318
Gender (male)	1.604	0.881 2.919	0.122
**Comorbidities**			
Pre-existing renal failure	1.268	0.672 2.394	0.464
Hypertension	1.101	0.561 2.161	0.780
Diabetes mellitus	0.698	0.368 1.326	0.272
**Laboratory findings**			
Creatinine > 100 μmol/L	1.358	0.724 2.547	0.340
CRP > 100 mg/L	1.294	0.723 2.317	0.385
Lymphopenia < 500/mm^3^	1.955	1.027 3.723	0.041 *
Neutrophils > 10000/mm^3^	1.674	0.753 3.722	0.206
Eosinophils = 0/mm^3^	0.892	0.504 1.580	0.696
Delta Eosinophils + (H-24)	0.696	0.391 1.239	0.218

Abbreviations: BMI = body mass index, CRP = C reactive protein, Delta+ = positive difference between circulating eosinophils rate at admission to the ED, and after 24 hours of hospital stay (H-24). * *p* < 0.05.

## Data Availability

All data analyzed as part of the study are included.

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
