# Peer review of "Do Blood Eosinophils Predict in-Hospital Mortality or Severity of Disease in SARS-CoV-2 Infection? A Retrospective Multicenter Study"

_microorganisms, 2021, doi:10.3390/microorganisms9020334_

Round 1

Reviewer 1 Report

Interesting report with potential impact on the clinical practice. The data are presented in clear way; however, I have two comments:

  1. Despite the fact, that eosinopenia was not predictive for the mortality, still there are some results (despite the low area under RIC curve). However, the title of the paper is too strong. Still I feel that there is a potential to predict at least the severity, especially by repeated evaluation of EOS during the course of the diseases. Authors discussed the limitation and potential of their findings. Absolute eosinopenia showed some capacity to predict the severity.
  2. Eosinopenia is still useful marker in the combination with the other biomarkers, parameters and clinical symptoms, especially at the initial evaluation of the suspected COVID-19. 

Author Response

Foremost, we are grateful for the opportunity provided to share our research work, and humbled by the pertinent comments presented by the reviewer 1.

  1. We have modified our title and conclusions to be less absolute, and therefore more in tune with our final results which, indeed, show eosinopenia’s potential to predict severity.
  2. We also agree with this point. Eosinopenia is almost permanent in cases of SARS-CoV-2 infection; it does not seem to a sufficient predictor by itself, yet in combination with other biomarkers (e.g. inflammation parameters) and clinical symptoms (e. g. fever), eosinophils might be of interest.

Reviewer 2 Report

This is an interesting study about eosinopenia as a predictor of COVID-19 severity and mortality. The authors found that eosinopenia was not a good predictor in COVID-19 at multivariable analysis. They collected data from 1,035 patients, with a confirmed diagnosis of moderate or severe COVID-19.

The paper is well written. However, some issues remain.

In the Methods section, the authors must report eosinophils cutoff that was used as definition of eosinopenia.

I think that statistic analyses should be performed also including patients with mild COVI-19 who underwent ambulatory care, in order to compare to results obtained from only patients who were hospitalized.

Since undetectable eosinophils represented a prognostic factor for COVID-19 severity, title and conclusion in the abstract and the text should be corrected or better described.

Moreover, the authors should explain why did not perform multimarkers analyses, such as suggested in the discussion.

The discussion section is well written and limitations of the study are reported.

Author Response

In light of improving our work, we also thank reviewer 2 for their remarks.

  1. As requested, we have added eosinophils cutoff that was used as definition of eosinopenia in the Methods section.
  2. Concerning patients with mild COVID-19 who underwent ambulatory care, it would have, indeed, been interesting to include them in order to better analyze and compare eosinophils in 3 sub-populations of different severity. However, we could not obtain these data, mainly because the majority of these ambulatory patients did not have any blood test or CBC analysis.
  3. We have modified and softened the title and conclusions of our manuscript according to our findings.
  4. A multi-markers analysis is, indeed, suggested in the Discussion section. However, this study focuses solely on the performance of eosinopenia. A multi-marker approach could only be suggested to be the right formula as it requires a much more in-depth analysis that may be biased by the fact that our study was designed to only study eosinopenia (exclusion criteria in particular).

Reviewer 3 Report

In the publication, the authors showed that eosinophil counts are not a sensitive marker for predicting the severity of COVID-19.

Please find below my comments:
1. literature for sentence L43-44 should be added. The number of patients should be updated.
2. Please add this manuscript to sentence L50-52 (doi: 10.3390/pathogens9060493).
3. The purpose of the work has been clearly defined and does not raise any objections.
4. in the methodology, please add the definitions of comorbidities (authors can also add these definitions as supplementary materials).
5. caption for figure 2 - it makes no sense to explain the ICU abbreviation once again here.
6. the methodology does not raise any objections.
7. the results are presented in a clear manner and also do not raise objections to the reviewer.
8. the discussion is written in an interesting way.

Author Response

We also thank reviewer 3 for their constructive feedback and have responded point by point to the comments:

  1. Literature has been added for L43-44 and we have updated the figures mentioned according to WHO.
  2. Reference indicated for the L50-52 has also been added.
  3. We are grateful for the comment.
  4. In the Methods section, we detailed co-morbidities as requested. However, due to the fact that we faced an overwhelming crisis situation during the peak of the outbreak, it was quite difficulty to collect detailed data. For example, in the case of history of cardiovascular disease, local investigators simply identified then recorded whether or not the patient had pre-existing cardiovascular disease (all causes combined). We did not estimate it usefull to be more specific, especially as this is not related to our main research question.
  5. We have removed the repetitive abbreviation as requested.

6-8. We are delighted to meet your standards, thank you again.

Round 2

Reviewer 2 Report

Thank you for improving the paper and softening conclusions.